# Combined Inhibition of FOSL-1 and YAP Using siRNA-Lipoplexes Reduces the Growth of Pancreatic Tumor

**DOI:** 10.3390/cancers14133102

**Published:** 2022-06-24

**Authors:** Lara Diego-González, Andrea Fernández-Carrera, Ana Igea, Amparo Martínez-Pérez, M. Elisabete C. D. Real Oliveira, Andreia C. Gomes, Carmen Guerra, Mariano Barbacid, África González-Fernández, Rosana Simón-Vázquez

**Affiliations:** 1CINBIO, Universidade de Vigo, Immunology Group, 36310 Vigo, Spain; ldiego@uvigo.es (L.D.-G.); andfernandez@uvigo.es (A.F.-C.); aigea@uvigo.es (A.I.); ammartinez@uvigo.es (A.M.-P.); africa@uvigo.es (Á.G.-F.); 2Instituto de Investigación Sanitaria Galicia Sur (IIS Galicia Sur), SERGAS-UVIGO, 36312 Vigo, Spain; 3CFUM (Center of Physics), Department of Physics, University of Minho, Campus of Gualtar, 4710-057 Braga, Portugal; beta@fisica.uminho.pt; 4CBMA (Centre of Molecular and Environmental Biology), Department of Biology, University of Minho, Campus of Gualtar, 4710-057 Braga, Portugal; agomes@bio.uminho.pt; 5CNIO (Centro Nacional de Investigaciones Oncológicas), Experimental Oncology Group, 28029 Madrid, Spain; mcguerra@cnio.es (C.G.); mbarbacid@cnio.es (M.B.); 6Centro de Investigación Biomédica en Red de Cáncer (CIBERONC), Instituto de Salud Carlos III, 28029 Madrid, Spain

**Keywords:** pancreatic ductal adenocarcinoma, nanomedicine, liposomes, gene silencing, KRAS, Hippo pathway

## Abstract

**Simple Summary:**

Intercepting the molecular mechanisms implicated in pancreatic cancer progression can be an efficient therapeutic approach to treat this aggressive tumor. The Hippo pathway is a key mechanism driving tumor progression, even in the absence of KRAS activation. When this pathway is switched off, the transcriptional coactivator YAP is translocated into the nucleus and induces the activation of several genes implicated in tumor progression and apoptosis inhibition. FOSL-1 is a transcription factor that synergizes with YAP, forming a transcriptional complex. It has been shown to have a good therapeutic outcome when they are individually inhibited. In this work, we showed for the first time that the combined inhibition of YAP and FOSL-1 mRNA expression, using siRNA-lipoplexes, induces superior control over tumor growth in vitro and in vivo, compared to the individual treatments, and a reduction of the tumor stroma. The results offer a new therapeutic approach for pancreatic cancer treatment.

**Abstract:**

Pancreatic cancer evades most of the current therapies and there is an urgent need for new treatments that could efficiently eliminate this aggressive tumor, such as the blocking of routes driving cell proliferation. In this work, we propose the use of small interfering RNA (siRNA) to inhibit the combined expression of FOSL-1 and YAP, two signaling proteins related with tumor cell proliferation and survival. To improve the efficacy of cell transfection, DODAB:MO (1:2) liposomes were used as siRNA nanocarriers, forming a complex denominated siRNA-lipoplexes. Liposomes and lipoplexes (carrying two siRNA for each targeted protein, or the combination of four siRNAs) were physico-chemically and biologically characterized. They showed very good biocompatibility and stability. The efficient targeting of FOSL-1 and YAP expression at both mRNA and protein levels was first proved in vitro using mouse pancreatic tumoral cell lines (KRAS^G12V^ and p53 knockout), followed by in vivo studies using subcutaneous allografts on mice. The peri-tumoral injection of lipoplexes lead to a significant decrease in the tumor growth in both Athymic Nude-Foxn1^nu^ and C57BL/6 mice, mainly in those receiving the combination of four siRNAs, targeting both YAP and FOSL-1. These results open a new perspective to overcome the fast tumor progression in pancreatic cancer.

## 1. Introduction

Pancreatic ductal adenocarcinoma (PDAC) is the most aggressive and lethal cancer [1]. This type of tumor grows rapidly and due to its late diagnosis and resistance to treatment, most patients have a poor prognosis, with a five-year survival rate of 6–10% [2,3].

Currently, surgical resection combined with chemotherapy including gemcitabine, nab-paclitaxel or FOLFIRINOX (Folinic acid, 5-FU, Irinotecan and Oxaliplatin), and immunotherapy, are the available therapies to improve patient survival [4]. However, their efficacy is very limited and there is an urgent need for new therapies.

It is known that approximately 90% of patients with PDAC are characterized by a mutation in the KRAS oncogene, generally in codon 12 [5]. As a result, KRAS protein is constantly activating downstream signaling pathways which are involved in cell progression, survival and metastasis [6]. Likewise, recent studies have shown that the transcriptional coactivator Yes associated protein (YAP) and its paralog with a PDZ-binding motif TAZ are activated downstream to KRAS in the development of PDAC [7,8]. YAP/TAZ are both involved in the Hippo pathway [7]. When the Hippo pathway is switched off, YAP and TAZ can be translocated to the nucleus, where they bind to DNA-binding factors of the transcription enhancer factor (TEA)-domain (TEAD) family, to regulate gene expression [9]. Thus, high levels of YAP protein in cancer cells has been associated with promotion of tumor proliferation [10], and its suppression could be a promising therapy [11,12,13,14]. Besides, YAP activation can even circumvent the need of KRAS activation in PDAC after the induced extinction of KRAS signaling [15]. For that reason, YAP could be a relevant target in both KRAS-mutated and KRAS-independent PDACs.

FOSL-1, also known as FOS-related antigen 1 (FRA1), is a transcription factor that synergizes with YAP and has been associated to poor prognosis in PDAC patients [7]. It belongs to the Activator Protein-1 (AP-1) family of transcription factors [16], which stimulate proliferation and metastasis in different tumors, such as in pancreatic cancer, thanks to its high expression [17]. In recent years, it has been defined as a potential therapeutic target to stop tumor growth [18,19,20].

Gene therapy is a great alternative for the treatment of those diseases that do not have efficient chemotherapeutic drugs [21,22]. The use of RNA interference mechanisms (RNAi) and, in particular, small interference RNA (siRNA), allows a reversible inhibition of gene expression without permanent genome modification [23,24,25]. These small regulatory nucleic acids are trapped by the RNA-induced silencing (RISC) complex when they form a double stranded RNA (dsRNA) structure. The RISC complex facilitates the binding of the regulatory single stranded RNA (ssRNA) to a complementary messenger RNA (mRNA) sequence, through base-pairing interactions, inducing the silencing of the protein expression by several different mechanisms, such as mRNA degradation [26].

Despite its advantages, the in vivo use of siRNA is limited due to its high instability and the need of transfection into the target cells [27,28,29]. Essentially, siRNA is susceptible to rapid degradation by serum endonucleases and metalloproteases, culminating in their glomerular filtration and rapid excretion by the kidneys [24,30]. Another challenge in transporting these nucleic acids to the cells is the extravasation from the blood vessels into the tissues [29,31].

Nanocarriers are used to overcome these disadvantages. In the oncological field, lipid-based nanostructures, such as liposomes, are commonly used due to their high biocompatibility and versatility to transport different type of molecules [32]. In 1987 Felgner et al. proposed cationic liposomes as nucleic acid carriers [33]. In this work, we used cationic liposomes based on the cationic lipid dioctadecyldimethylammonium bromide (DODAB) and the neutral lipid 1-monooleoyl- rac-glycerol (MO) that were first proposed by Real Oliveira et al. in 2010 [34,35] for nucleic acid transport, and later on validated for the administration of siRNA [36,37,38].

Our hypothesis is based on the potential therapeutic effect of the combined blockade of FOSL-1 and YAP expression, avoiding their synergistic effect in the promotion of PDAC survival and proliferation. In this study, we propose the use of a combination of two different specific siRNA targeting FOSL-1 and two specific siRNA targeting YAP, encapsulated into DODAB:MO liposomes to achieve the downregulation of these signaling proteins. The therapeutic efficacy of the siRNA-lipoplexes was tested in two different mouse pancreatic ATQ314G and ATQ303G cell lines (KRAS^G12V^ and p53 knockout), and in a subcutaneous allograft of ATQ303G in Athymic Nude-Foxn1nu and C57BL/6 mice.

## 2. Materials and Methods

### 2.1. Preparation of Liposomes

DOBAB and MO lipids were purchased from TCI Chemicals (Zwijndrecht, Belgium) and (Sigma-Aldrich San Luis, MO, USA), respectively. DODAB:MO cationic liposomes (molar ratio 1:2) were synthesized by film re-hydration followed by extrusion. Briefly, well-defined volumes of the lipids (20 mM in ethanol) for a final concentration of 3 mM were added to a rounded tube and exposed to vacuum in a bath at 60 °C for 10 min to evaporate the solvent. The formed lipid film was re-hydrated with 5 mL of endotoxin-free water for 15 min at 60 °C under rotation. Finally, to achieve a homogeneous population, the resultant liposomes were extruded 2 and 6 times at 60 °C using 400 and 100 nm filters (Whatman^®^, Maidstone, UK), respectively. The liposomes were stabilized at room temperature (RT) for 1 h before use.

For cell internalization studies, the nanostructures were labeled with rhodamine B by adding L-α-Phosphatidylethanolamine-N-(lysamine rhodamine B sulfonyl) (Avanti Lipid Polar, Birmingham, AL, USA) (25 μL at 3 mM in absolute ethanol) into the lipid mixture at a molar ratio of 1:200 (rhodamine:lipids) and following the same protocol described for the synthesis of liposomes. To avoid the degradation of the rhodamine, the sample was prepared and stored in dark conditions.

### 2.2. Preparation of Lipoplexes

Five complementary double stranded siRNAs (Table 1), two for each target protein (FOSL-1 and YAP) and an irrelevant one, were designed by Ambion (Thermo Fisher Scientific, MA, USA) and synthesized by IDT (Integrated DNA Technologies, Inc., Iowa, USA). They were dissolved in RNA-free water (stock at 1 mM), under aseptic and cold conditions to avoid any contamination or degradation.

For the experiments, two specific siRNAs (_1+2_) for each protein and combinations targeting FOSL-1_1+2_ and YAP_1+2_ were tested. To obtain the siRNA-lipoplexes (from here called lipoplexes), 2 µM of siRNA (Fosl_1+2_, Yap_1+2_ or Fosl_1+2_ + Yap_1+2_), diluted in HEPES buffer, 25 mM pH 7.4, and liposomes were warmed in a water bath at 60 °C for 10 min. Then, 100 µL of siRNA were mixed with different amounts of liposomes depending on the specific charge ratio (±) (5, 10, 15 or 20) [± = positive moles (DODAB)/negative moles (phosphate of siRNA)]. After a short incubation at 60 °C and homogenization in the vortex, the lipoplexes were maintained at RT for 20 min before their use.

### 2.3. Determination of siRNA Encapsulation

#### 2.3.1. Agarose Gel Electrophoresis

For the determination of a correct encapsulation of the siRNA inside the liposomes, an agarose gel electrophoresis was carried out. The lipoplexes, as well as liposomes and free siRNA as controls, were loaded into a 2% agarose gel in TAE 1X (Omega Bio-Tek, Georgia, USA). The samples loaded with SYRB Green (Invitrogen™, MA, USA) were exposed to electrophoresis at 50 V for 10 min. The gel was read in the equipment ChemiDoc™ XRS+ Imaging Systems with the Image Lab™ software (Bio-Rad Laboratories, Inc., CA, USA).

#### 2.3.2. Quant-it™ RiboGreen RNA Assay Kit

The Quant-it™ RiboGreen RNA Assay Kit (Invitrogen™, MA, USA), a fluorescent stain for quantitating RNA in solution, was used to corroborate and quantify the encapsulation of siRNA into DODAB:MO liposomes. Briefly, the reagent was diluted at 2X in TE Buffer, RNase-free (200 mM Tris-HCl, 20 mM EDTA, pH 7.5 in DEPC-treated water). On 96-well black bottom-clear plates (Invitrogen™, MA, USA), 100 µL of lipoplexes at different charge ratios (±) plus 100 µL of RNA reagent at 2× were incubated for 5 min in the dark. Free siRNA and empty liposomes, at the same concentration used in the synthesis of the lipoplexes, were also tested for comparison.

Fluorescence (F) (excitation/emission: 485/538 nm) was measured on an EnVision multidetector (Perkin-Elmer Inc., Norwalk, CT, USA). The percentage of encapsulation efficiency (% Encapsulation) was determined as follows:$$\%\;Encapsulation=\left(\frac{1-F\left(lipoplexes\right)}{F\left(siRNA\right)}\right)\times 100$$

F(lipoplexes): fluorescence of non-encapsulated siRNA; F(siRNA): fluorescence of free siRNA.

### 2.4. Physical-Chemical Characterization

Both liposomes and lipoplexes were characterized by determining mean size, polydispersity index (PDI) and zeta potential (ζ-potential) using DLS technology (Zetasizer^®^, NanoZS, Malvern Instruments, Malvern, UK). Samples were diluted 1:12 in endotoxin-free water or PBS in a final volume of 1 mL at 25 °C to study the mean size and PDI. In the case of ζ-potential, the solvent was potassium chloride (KCl) at 1 mM. Three measurements were made for each parameter.

### 2.5. Sterility

During the synthesis of liposomes and lipoplexes, sterile conditions were used at all the steps, when possible. Moreover, to discard the presence of potential microbial contaminations that could interfere with the biological response, both liposomes and lipoplexes, were tested to confirm the absence of endotoxin or bacteria.

The presence of endotoxin was evaluated by the Gel Clot kit following manufacture protocol (Associates of Cape Cod, Inc., MA, USA). The result is considered positive when a compact gel clot is formed, otherwise is negative (or lower than 0.03 EU/mL, limit of detection).

To test the potential bacterial contamination, liposomes were seeded in LB agar LENNOX plates (Condalab, Madrid, Spain) under aseptic conditions and kept in an incubator at 37 °C. The absence or presence of bacterial colonies was monitored daily (24–72 h).

### 2.6. In Vitro Studies

#### 2.6.1. Cell Lines

The mouse pancreatic (ATQ314G and ATQ303G) tumor cell lines used in this study were generated from PDAC tumors developed by Elastase-tTA; Tet-O-Cre; K-Ras^+/LSLG12Vgeo^; p53^lox/lox^ mice (C57BL6/129 background), as described by C. Guerra et al., 2011 [39].

The THP-1 and MIA-PaCA-2 cell lines were purchased from the American Type Culture Collection (ATCC) (Virginia, USA), and the human peripheral blood mononuclear cells (hPBMCs) were obtained from three healthy donors by density gradient centrifugation (Ficoll, GE Healthcare, Boston, MA, USA) from whole blood. The hPBMCs were recovered, washed with PBS, and kept in culture medium.

The cultures medium used were DMEM Glutamax for mouse pancreatic cell lines, DMEM for MIA-PACA-2 and RPMI 1640 for THP-1 and hPBMCs. All culture mediums were supplemented with 10% of fetal bovine serum (FBS) and 2% of penicillin/streptomycin) (Gibco™, Thermo Fisher Scientific, MA, USA).

Cells were kept at 37 °C and under 5% CO_2_, and their maintenance was performed every two or three days, when the cell confluence achieved 70–80%.

#### 2.6.2. Cellular Uptake

The internalization of rhodamine-labeled liposomes by mouse pancreatic tumor cells was determined by flow cytometry. Cells were seeded at a density of 5 × 10^4^ cells/well on 96-well plates (COSTAR^®^, Corning Inc., New York, NY, USA) and rhodamine-labeled-liposomes at 100 µM were added and incubated at different time points (30 min, 1, 3 and 5 h). Then, cells were washed with PBS and detached with Trypsin-EDTA 1X (Sigma-Aldrich, San Luis, Missouri, USA). The action of trypsin was neutralized with complete culture medium and cells centrifuged at 1200 rpm for 5 min and washed with PBS twice. Finally, cells were resuspended in 200 μL of PBS and analyzed on the Cytoflex S cytometer (Beckman Coulter, CA, USA) using the FL3 channel. The final data was represented as median fluorescence intensity (MFI).

#### 2.6.3. Cell Viability in the Presence of Liposomes and Lipoplexes

Three different methods were used: MTS colorimetric assay, impedance measurement using the xCELLigence^®^ system and cellular apoptosis by flow cytometry. In all cases, lipoplexes were tested at 50 nM of total siRNA and liposomes were added at 100 µM (equivalent of 50 nM siRNA) or at 50 µM in FBS-free medium. Untreated cells and culture medium were used as controls.

In the case of MTS, MIA-PACA-2 and macrophage-differentiated THP-1 cells were seeded at 6 × 10^3^ and 1.5 × 10^4^ cells/well on 96-well plates. The next day, the treatments were added, and after 4 h of incubation, the medium was removed and renewed by complete medium. 72 h later, the MTS reagent was dissolved and added according to the manufacturer’s instructions (BioVision Incorporated, Milpitas, CA, USA). Finally, the absorbance at 490 nm was measured in the EnVision multidetector. The percentage of cell viability was calculated following the formula below:$$Cell\;viability\;\left(\%\right)=\frac{Abs\;\left(Cs+Lipos\right)-Abs\;Lipos\;}{Abs\;Cs-Abs\;culture\;medium}\times 100$$

The second method included the growing of cells in special plates, measuring the impedance with the xCELLigence^®^ RTCA DP Instrument (RocheDiagnostics, Mannheim, Germany), which allows real-time analysis with data recorded every 15 min. Both mouse pancreatic cancer cells (ATQ314G and ATQ303G) were seeded at a density of 7.5 × 10^3^ cells/well. When cells reached the exponential phase of growth, lipoplexes were added in FBS free medium. After 4 h of incubation, the medium was renewed by complete culture medium. After 96 h of incubation, the cell index was normalized respect to the moment of adding the treatments. The percentage of cell viability was normalized respect to the untreated cells at different time points (24, 48, 72 and 96 h).

Cytotoxicity induced by the lipoplexes was also studied by a double labeling with Annexin V-FITC and Propidium Iodide (PI) (Immunostep S.L, Salamanca, Spain), using flow cytometry. ATQ314G and ATQ303G cell lines were seeded on 48-well plates (COSTAR^®^, Corning Inc., New York, USA) at a density of 6 × 10^3^ cells/well. After 24 h of resting, lipoplexes were added. Like in other assays, culture medium was changed 4 h later. After 72 h of incubation, cells were washed with PBS, labelled with Annexin V-FITC and PI following the manufacturer’s instructions, and analyzed by flow cytometry (Beckman Coulter FC500, CA, USA).

#### 2.6.4. Activation of the Complement Cascade

The activation of the complement system was determined by the degradation of factor C3 by Western blot in a pool of human plasma from healthy donors as previously described by us [40]. Liposomes were tested at three different concentrations (100, 50 and 25 µM), while lipoplexes were used at a final concentration of 50 nM of siRNA. PBS and Zymosan at 1 mg/mL (Sigma-Aldrich, San Luis, MO, USA) were used as negative and positive controls, respectively.

#### 2.6.5. Hemolysis

Whole blood was obtained from healthy mice by cardiac puncture with heparin as an anticoagulant. PBS was added in a 3% w/v ratio and then 80 μL of erythrocyte suspension was added in a 96 U-bottom plate (COSTAR^®^, Corning Inc., New York, NY, USA); 80 μL of lipoplexes or liposomes were added to a final concentration of 50 nM of siRNAs or equivalent. Plate was incubated 4 h at 37 °C, followed by centrifugation at 1000× *g* for 10 min at 4 °C and 80 μL of the supernatant were transferred to a new plate of 96 flat bottom wells. Absorbance was read at 558 nm on the EnVision multidetector. As positive and negative control, PBS and 1% Triton X-100 (Sigma-Aldrich, San Luis, MO, USA) in PBS were used, respectively. The percentages of hemolysis were calculated using the formula:$$Hemolysis\;\left(\%\right)=\frac{Abs\;Sample-Abs\;PBS}{Abs\;Triton-Abs\;PBS}\times 100$$

We followed the ASTM International protocol E2524-08 [41], which considers: 0–2% non-hemolytic, 2–5% moderately hemolytic and >5% hemolytic.

#### 2.6.6. FOSL-1 and YAP Silencing

The expression of FOSL-1 and YAP in ATQ314G and ATQ303G cell lines was determined by qRT-PCR and Western blot. 7.5 × 10^4^ cells were seeded on 60 mm culture plates (Falcon^®^, Corning Inc., New York, NY, USA) until they reached a confluence of 10–15%, when the culture medium was carefully removed and each type of lipoplexes (carrying 50 nM of siRNA) were added in FBS-free culture medium. After 4 h of incubation, the medium was changed by complete culture medium and cells were kept in culture for 72 h. Afterwards, cells were recovered.

Quantitative real-time PCR (qRT-PCR) by determining delta-delta Ct (ΔΔCt) was used to study the knock-down expression of FOSL-1 and YAP. RNA was purified with PureLink^®^ RNA Mini Kit (Invitrogen, MA, USA). For the synthesis of cDNA, the Superscript II Reverse Transcriptase kit (Invitrogen, MA, USA) was employed. RNA and cDNA were quantified using the NanoDrop 2000c. cDNA was used for qRT-PCR using SYBR Green Master Mix (Applied Biosystems, MA, USA) in the 7900HT Fast Real Time PCR System (Applied Biosystems, MA, USA). The conditions used for the PCR were: 50 °C (2 min) and 95 °C (10 min), followed by 40 cycles at 95 °C (15 seg), 60 °C (30 seg) and 72 °C (30 seg), using the primers indicated in Table 1. Threshold cycle (Ct) values were given automatically by the program. The data were normalized respect to the internal control Glyceraldehyde-3-Phosphate Dehydrogenase (GAPDH), and the Ct was determined with respect to the untreated cells following to 2^−ΔΔCt^.
ΔΔCt = [(Ct target gene − Ct GAPDH) treated cells] − [(Ct target gene − Ct GAPDH) untreated cells].

For protein silencing, cells were lysed using the radioimmunoprecipitation assay (RIPA) buffer and the quantification of the proteins was carried out using the kit Bio-Rad proteins (Bio-Rad Laboratories, Inc., CA, USA) following manufacturer’s instructions, followed by analysis in the NanoDrop 2000c (Thermo Scientific, MA, USA). Finally, 40 µg of protein was loaded into a 10% SDS-PAGE gel and transferred to a nitrocellulose membrane using the Transblot Semidry Transfer Equipment (Bio-Rad Laboratories, Inc., CA, USA). The membranes were blocked with 5% nonfat milk (Sigma-Aldrich, San Luis, MO, USA) in PBS-T for 1 h at RT and incubated overnight at 4 °C with primary antibodies diluted 1:500 in 2.5% bovine serum albumin (BSA) (VWR Chemicals, Pennsylvania, USA) in PBS-T. The anti-mouse IgG1 against Fra-1 (D-3) and IgG2a against Yap (63.7) monoclonal antibodies were both from Santa Cruz Biotechnology (Inc., Texas, USA). Tubulin was chosen as the internal control and the monoclonal antibody β-tubulin (BT7R) (Invitrogen, MA, USA) used for its detection. The membranes were incubated with the secondary anti-mouse IgG antibodies conjugated with horseradish peroxidase (HRP) (Bio-Rad Laboratories, Inc., CA, USA) diluted 1:5000 in 2.5% BSA in PBS-T. After a washing process, the protein bands were revealed by adding Clarity substrate solution (Bio-Rad Laboratories, Inc., CA, USA) that reacts with HRP. The intensity of the bands was quantified using ChemiDoc™ XRS+ Imaging Systems with image Lab™ software. The target protein/tubulin ratio was normalized with respect to the negative control.

### 2.7. In Vivo Studies

Seven-week-old female Athymic Nude-Foxn1^nu^ and C57BL/6 mice were purchased from Envigo (Envigo RMS, Spain S.L). The mice were kept under aseptic conditions at the specific-pathogen-free (SPF) animal house facilities at the University of Vigo, Spain. Animals were fed *ad libitum* with sanitary commercial mouse diet under uniform light (12 h light/dark periods), temperature and humidity (25 ± 1 °C and 50 ± 5%, respectively) controlled conditions. Throughout the experiments, the animals were monitored daily, in order to analyze evolution of tumor growth, clinical symptoms or illness.

Nude and C57BL/6 mice (8 per group) were divided into five groups (PBS, irrelevant lipoplexes, Fosl_1+2_ lipoplexes, Yap_1+2_ lipoplexes and Fosl_1+2_ + Yap_1+2_ lipoplexes). After a preliminary experiment to optimize the cell number, the subcutaneous allograft model was generated by injection of 1 × 10^6^ (nude) or 3 × 10^6^ (C57BL/6) ATQ303G cells in cold 100 µL of PBS-Matrigel (1:1) (Corning^®^ Matrigel^®^, New York, NY, USA) at the right flank of each mouse. Tumor growth was daily monitored, and treatments were initiated when tumors reached a volume of 50–100 mm^3^. Lipoplexes loaded with 2 µg of total siRNA were injected subcutaneously around the tumor. In the C57BL/6 mice, 4 µg of total siRNA was also used for the Fosl_1+2_ + Yap_1+2_ lipoplexes. As controls, PBS and lipoplexes with unspecific siRNA were used. The treatments were injected four times (48 h between doses). Mice were sacrificed by CO_2_ three or two days after the last injection (for nude or C57BL/6 mice, respectively). Tumors were recollected and half of the tumors were used for histological study, and the other half were stored at −80 °C for subsequent expression analysis of FOSL-1 and YAP mRNA.

#### 2.7.1. qRT-PCR to Analyze the Silencing of FOSL-1 and YAP in the Tumors

The analysis of the knock-down expression of FOSL-1 and YAP in the tumor was performed using qRT-PCR by determining ΔΔCt. Tumor RNA was extracted using the RNA-Solv^®^ Reagent (Omega, BIO-TEK, Norcross, Georgia, USA) following the manufacturer’s instructions. The following steps are identical to those described in the in vitro section.

#### 2.7.2. Histological Studies

Tumor samples were fixed in 4% formalin (Sigma-Aldrich, San Luis, Missouri, USA) for 24 h and then transferred to 70% ethanol until further processing. Paraffin block formation was carried out following the standardized protocol using the Inclusion center with cold plate Leica EG1150H (Leica Biosystems Nussloch GmbH, Nussloch, Germany). For Trichrome staining (Sigma-Aldrich, San Luis, MO, USA), tissue was sectioned by Rotational microtome Leica RM2255 at 7 µm (Leica Biosystems Nussloch GmbH, Nussloch, Germany), mounted onto glass slides and dried at 37 °C at least for 24 h. Desparaffinization process and the subsequent stainings were also carried out according to the manufacturer’s instructions. A blind analysis of the tumor slides was performed by an expert in clinical pathology anatomy from the Hospital Meixoeiro (Vigo). Images were obtained with a direct Nikon Eclipse (Nikon, Tokyo, Japan) microscope and the software NIS-Elements D 4.30.02.

#### 2.7.3. Statistical Analysis

The results were represented as the mean ± standard deviation (SD) of three independent experiments (three replications per experiment), except to the in vivo tumor volume data, being expressed as the mean ± standard error of the mean (SEM). Shapiro-Wilk and Kolmogorov-Smirnov tests were carried out to determine the distribution of the samples. T-student or Mann Whitney Wilcoxon tests were used to determine significant differences between treatments and for the in vitro experiments. In the case of animal studies, Kruskal Wallis or Anova test on the GraphPad Prism 8 software (GraphPad Software, Inc., La Jolla, CA, USA) was used. Differences were considered statistically significant when the probability (P) was lower than 0.05 (*p* ≤ 0.05). In the figures, the statistically significant results are referred to as: * *p* ≤ 0.05, ** *p* ≤ 0.01, *** *p* ≤ 0.001, **** *p* ≤ 0.0001.

## 3. Results and Discussion

### 3.1. Physicochemical Characterization and Stability of Liposomes and Lipoplexes

DODAB:MO (1:2) liposomes were synthesized by the Bangham’s method, which consist of a lipid film hydration followed by extrusion, and the encapsulation of siRNA was carried out by electrostatic interactions. Their physicochemical characterization (mean size, PDI, and ζ-potential) is shown in Table 2. Liposomes showed a homogeneous population (PDI < 0.2) with a mean size around 140 nm and a positive surface charge of 51 mV. Its cationic character is able to promote the interaction and encapsulation of siRNAs as well as its contact and uptake by cells.

The stability of the liposomes was followed over the time, being very stable at least during 2 months at 4 °C in water (Figure 1A). When liposomes were in PBS with 10% FBS, the presence of salts and serum induced an increase in both mean size and PDI. However, the opposite occurs with their surface charge, decreasing in the presence of PBS-FBS, but maintained stability for at least 24 h (Figure 1B). Thus, the presence of salts and proteins leads to increase the size of the liposomes, masking the DODAB molecules on the surface, which translates into a decrease in their positive charge.

Lipoplexes were formed by mixing the liposomes and the designed siRNAs: two to target FOSL-1 (Fosl_1+2_), two for YAP (Yap_1+2_) or their combination (Fosl_1+2_ + Yap_1+2_) (siRNA sequences are in Table 1). The formation of lipoplexes is based on the charge ratio (±) which is defined as the ratio between the positive charge of liposomes provided by the cationic lipid DODAB and the negative charge of the phosphate groups in the RNA.

We evaluated the correct encapsulation of the siRNA at different ratios (from 5–20) by electrophoresis of agarose gel, not observing free RNA in the lipoplexes at any ratio studied (Figure 2A). This data was corroborated using another technique, the RiboGreen RNA Assay Kit (Figure 2B). Although there were not significant differences in the encapsulation efficiency, the charge ratios 15 and 20 were considered the best, guaranteeing total incorporation of the siRNA, in agreement with previous experiments by Oliveira A. C. N. et al., [36]. Hence, based on these results and taking into account the concentration of liposomes necessary for the formation of lipoplexes, the charge ratio selected for the rest of the experiments was 15. As shown in the Figure 2C, with this ratio we obtained a good encapsulation efficiency using the different combinations of siRNA.

The total siRNA concentration was the same in all lipoplexes. Thus, in those carrying four different siRNAs (like Fosl_1+2_ + Yap_1+2_), the concentration of each siRNA is half of that one used for those targeting FOSL-1 (Fosl_1+2_) or YAP (Yap_1+2_).

As seen in Table 2, lipoplexes maintained both a low PDI and a similar size than those in liposomes. The greatest difference was observed in terms of surface charge. When we encapsulated more than one siRNA, especially with four siRNAs, the ζ-potential decreased slightly, indicating that some molecules could be adsorbed onto the liposome surface instead of being all encapsulated. Besides, the liposomes could undergo some small lipid rearrangement on the surface to accommodate the siRNA, placing some DODAB molecules in the core and some MO molecules in the surface. Taken altogether, including the high efficiency of encapsulation and that the incorporation of siRNA does not seem to alter significantly the conformation of the liposomes, the results indicate that lipoplexes were successfully formed.

It is important to note that both liposomes and lipoplexes were synthesized under sterile conditions to avoid any potential microbiological contamination that was further confirmed by an endotoxin test and by seeding the samples in LB agar plates (see Section 2.5). In all cases, no endotoxin or bacterial contamination were detected in our samples (data not shown).

### 3.2. Cellular Uptake of Liposomes

It is important to verify if the pancreatic cancer ATQ314G and ATQ303G cells can capture liposomes and thereby allow lipoplexes to enter the cytoplasm, where they should exercise their targeting function. For this purpose, liposomes were labelled with rhodamine during their synthesis and then added to the cells. After 5 h this of incubation, cells were washed and analyzed by flow cytometry. Looking at the percentage of positive cells, both cell lines internalized the labeled liposomes very efficiently, with over 97% of positive cells (Figure 3A), although ATQ303G cells showed higher median fluorescence intensity (MFI) (Figure 3B), which could be correlated with a higher amount of internalized liposomes. We also tested shorter incubation times (30 min, 1 h and 3 h) in the ATQ314G cell line, and although internalization was already observed at 30 min, the labeling increased with longer exposure times.

### 3.3. FOSL-1 and YAP mRNA and Protein Silencing

DODAB:MO (1:2) lipoplexes carrying combinations of siRNA (Fosl_1+2_, Yap_1+2_ and Fosl_1+2_ + Yap_1+2_) at charge ratio (±) of 15 with 50 nM of siRNA, and incubation time of 72 h, were selected as the best conditions to study protein silencing in both tumoral pancreatic cell lines (ATQ314G and ATQ303G). The siRNA sequences were previously validated in both cell lines using lipofectamine as the transfection agent. For both proteins, a decrease of about 40–60% of the protein expression was achieved.

The inhibition of the expression at the mRNA level was studied by qRT-PCR. The results showed a decrease in the mRNA expression that was significant for the YAP protein in both pancreatic cancer cell lines incubated with the lipoplexes containing two specific siRNA for YAP (Yap_1+2_) (Figure 4A). FOSL-1 mRNA level was also inhibited in the ATQ314G cell line, compared to the untreated cells, although the inhibition was only significant for the lipoplexes carrying the four siRNAs, and showed a statistical tendency for the lipoplexes targeting FOSL-1 (*p* = 0.0519). On the contrary, ATQ303G cells showed no inhibition and higher variability.

The inhibition efficiency was also confirmed at protein level by Western Blot (Figure 4B and Appendix A). The results showed the same trend as those obtained by qRT-PCR. YAP protein was more inhibited than the FOSL-1 protein, either using lipoplexes carrying two siRNAs specific for YAP protein or with the combination targeting YAP and FOSL-1. A significant decrease in its expression levels was observed in both cell lines. Down-regulation was also observed for the FOSL-1 protein but did not reach statistical significance.

In summary, DODAB:MO (1:2) liposomes showed transfection efficiency, by means of the silencing of the mRNAs coding for YAP and FOSL-1, showing a decrease in both transcripts and protein levels.

### 3.4. Cytocompatibility of Liposomes and Therapeutic Effect of Lipoplexes

It has been described that liposomes are ideal nanocarriers due to its biocompatible and biodegradable behavior [32,42]. To confirm the safety profile of DODAB:MO (1:2) liposomes, a battery of different assays was carried out.

The cytocompatibility of the empty liposomes was determined by xCelligence assay using ATQ314G and ATQ303G cells. Both cell lines showed a cell viability greater than 80% in the presence of DODAB:MO (1:2) liposomes at 100 µM (Figure 5A). Cytotoxicity was also tested in other types of cells, like human peripheral blood mononuclear cells (hPBMCs), macrophage-differentiated THP-1 and MIA-PaCa-2. In all cases, a good cell viability (>80%) was observed after 24 h of incubation (Appendix A). Since cell viability did not decrease below 75% at any of the time points studied, it can be considered that the designed DODAB:MO (1:2) liposomes are not toxic [43]. These results are in agreement with those found in the literature with other lipid nanostructures. In 2017, Y. Xia et al. [44], tested polycation liposomes in OVCAR8/ADR cells during 48 h, showing similar results. J. Lee et al. [45], demonstrated the good compatibility of PEGylated DC-Chol/DOPE liposomes in SKOV3 cells, or Singh et al. [46], showed the safety profile of liposomes based on the Phospholipon 90G and Cholesterol (70:30 molar ratio) lipids in the MCF-7 cell line. Likewise, the small decrease in cell viability by DOBAB:MO liposomes has already been previously described in other cell lines (L929, 293 and C2C12) using a similar concentration as the one used here, but in that case with an inverted lipid molar ratio, that is, DOBAB:MO (2:1) [47].

The liposomes and lipoplexes also showed a good hemocompatibility because they did not induce hemolysis neither activation of the complement cascade (Appendix A). The complement system plays an important role in the innate response and it can have both a pro- as well as an anti-tumour role, depending on the type of cancer [48,49,50,51,52,53]. 

After discarding the toxicity induced by the carrier, we evaluated the potential therapeutic effect of the lipoplexes in the pancreatic tumor cell lines. The lipoplexes did not significantly affect the ATQ314G cells but induced a significant drop in the cell index on the ATQ303G cells (Figure 5A), associated to the YAP protein silencing.

Interestingly, although the FOSL-1 protein silencing was not significant on both cell lines at 72 h (Figure 4B), Fosl_1+2_ lipoplexes were also able to induce a significant reduction of the cell viability in ATQ303G cells. Besides, Fosl_1+2_ + Yap_1+2_ lipoplexes induced a higher antitumoral effect than those targeting only one protein. Hence, targeting FOSL-1 expression is an efficient therapy that synergizes with YAP silencing, although FOSL-1 protein degradation and recovery could have different kinetics than YAP protein.

To confirm these results, a complementary study was performed by flow cytometry to determine if lipoplexes were inducing apoptotic cell death. Cells were incubated 72 h with different lipoplexes and further labelled with Annexin V -FITC and Propidium Iodide (PI). The results are shown in Figure 5B. Apoptosis was observed in both cells lines, but especially in ATQ303G cells, confirming the potential therapeutic effect of lipoplexes. In addition, with the Yap_1+2_ and Fosl_1+2_ + Yap_1+2_ lipoplexes, the appearance of late apoptotic and necrotic populations was also observed in this cell line.

With these results, it is possible to conclude that a modulation in cell growth is achieved through the silencing of the expression of the FOSL-1 and YAP proteins in vitro. The ATQ303G cell line was more sensitive to this treatment in comparison to the ATQ314G cell line, likely related to the highest liposome uptake. For this reason, ATQ303G cells were selected to perform the in vivo studies.

### 3.5. Anti-Tumor Efficacy in Pancreatic Allografts in Immunosuppressed and Immunocompetent Mice

For the studies of therapeutic efficacy in animals, ATQ303G cells were injected into the right flank of Athymic Nude-Foxn1^nu^ and C57BL/6 mice. These cells were selected because of their better in vitro behavior in the presence of lipoplexes. As indicated in Figure 6A, the schedule of the study was as follows: once the tumor was established with a volume around 50–60 mm^3^, each animal received one peri-tumoral injection of lipoplexes, or PBS in the non-treated group, every two days, making a total of four injections per mouse. Figure 6B shows the normalized tumor volume in Athymic Nude-Foxn1^nu^ mice. The results show an evident reduction in tumor growth in the groups treated with the different lipoplexes, being significant for the group treated with the lipoplexes carrying the combination of siRNAs Fosl_1+2_ + Yap_1+2_. This result agrees with the in vitro experiments carried out in this cell line, where this combination induced the strongest mRNA and protein silencing.

It is worth noting that during the treatment and after the culling no signs of necrosis or inflammation were detected around the tumor or in the area of injection of the lipoplexes, in agreement with the cytocompatibility studies in vitro.

As previously mentioned, FOSL-1 and YAP proteins are related to tumor proliferation [8,54,55]. There are studies showing that both proteins cooperate to promote tumoral growth [8,56,57], which is corroborated by ours results. YAP and its homolog TAZ are the main effectors of the Hippo pathway, which is regulated by Mitogen-activated protein kinase (MEK), high extracellular matrix stiffness, G-protein-coupled receptors (GPCRs), cell cycle arrest or changes in the cell polarity [58]. After activation, YAP is translocated to the nucleus where it binds to the TEAD transcription factor family and induce the expression of genes implicated in proliferation, survival and cell migration [7]. FOSL-1 is one the genes induced by YAP/TEAD and a member of the activator protein 1 (AP-1) that synergizes and provides a positive feedback to YAP activation. In fact, FOSL-1 has been identified as part of the transactivation complex formed by JUN/FOSL-1, YAP/TEAD and ZEB1 (zinc finger E-box binding homeobox 1) that binds to the ADN to initiate the transcription of tumor-proliferation genes [59].

We observed that in the presence of lipoplexes combining two specific siRNAs against each protein, the tumor growth is hindered compared to the untreated mice. Although the silencing therapy alone failed to achieve complete tumor remission, a good control over tumor growth was observed during therapy administration. Therefore, the silencing of FOSL-1 and YAP could be an efficient adjuvant therapy to eradicate PDAC, avoiding tumor proliferation and probably metastasis.

In order to confirm that the lipoplexes were effectively targeting the expression of both FOSL-1 and YAP, we analyzed mouse tumor samples by qRT-PCR (Figure 6C). For both genes, a decrease in the expression levels was observed, albeit with high variability between mice. It is important to note that injections of lipoplexes were done around the tumor, not intratumorally, in order to manipulate it as little as possible. This could imply that lipoplexes may not reach all tumoral cells, especially those located inside. However, four injections of lipoplexes were enough to significantly avoid tumor growth.

A histological study was also carried out in the tumor samples (Figure 7), using Masson’s trichrome staining, to observe differences in the stroma and cell density. Control tumors (PBS and Irrelevant siRNA) presented a great tissue consistency, with an important amount of collagen (blue staining), while the treatments were able to induce a clear decrease of the tumor density and stroma, with a large number of dead cells.

In samples from those mice treated with the lipoplexes carrying two siRNAs specific for FOSL-1 or YAP (Fosl_1+2_ or Yap_1+2_), the cells presented a pink cytoplasm without nucleus, which correspond to apoptotic cells. These results are supported by our in vitro and in vivo data (Figure 5B and Figure 6), and a previous study using FOSL-1 protein inhibition, in which apoptosis was also found [55]. Interestingly, the tumors from animals treated with the combination of the four siRNAs (Fosl_1+2_ + Yap_1+2_) were smaller and their amount of collagen was reduced compared to control samples (Figure 7).

This effect is mostly related to the YAP function. In its inactive form, YAP is located in the cytoplasm and stabilized by several proteins, including the cytoplasmic actin and integrins that participate in focal adhesion of the cell to the extracellular matrix (ECM). The ECM stiffness of tumor stroma can activate YAP (though the activation of the focal adhesion integrins) that act as a mechanotransducer protein [60]. For that reason, the inhibition of YAP and its feedback protein FOSL-1 could have a strong influence on the tumor stroma in PDCA, in addition of inducing a significant inhibition of tumor growth due to their main role as transcriptional complex proteins of the Hippo pathway.

The anti-tumoral efficacy of the lipoplexes was also performed using immunocompetent C57BL/6 mice (Figure 8A–C). Mice showed higher variability in both the timing on the tumor growth and size reached, than on the SCID mice. This is mainly due to the immune response elicited against the C57BL6/129 mixed genetic background of the used tumor cell line. However, after the injection of lipoplexes, the results were very similar to those obtained in nude mice: a reduction in the tumor volume, especially when the four siRNAs are used. Hence, a competent immune system did not interfere the action induced by lipoplexes, neither enhancing nor inhibiting the silencing therapy. Conversely, the inhibition of YAP could be beneficial because it could avoid the immunosuppression induced by myeloid-derived suppressor cells (MDSCs), as observed in prostate adenocarcinoma in which the Hippo-YAP pathway is also activated [61]. 

In order to see if we could improve the anti-tumoral effect, we increased the dose using a twi-fold siRNA concentration (4 µg) of the Fosl_1+2_ + Yap_1+2_ lipoplexes, but similar tumor growth inhibition was obtained (data not shown), which indicates that the maximum therapeutic and synergistic effect can be achieved at lower doses when targeting both signaling proteins.

Interestingly, although the lipoplexes targeting FOSL-1 induced a lower inhibition of the mRNA and protein expression in vitro than those targeting YAP, the therapeutic efficacy in vivo seemed superior in the animals treated with the FOSL-1 lipoplexes (*p* = 0.0585 in C57BL/6 mice) compared to those treated with YAP lipoplexes alone (not a statistically significant difference). This could be due to the existence of other compensatory signaling mechanisms for YAP, such as the activation of its paralog TAZ [7]. Similarly, inhibition of other YAP-regulated genes, such as ribonucleotide reductase regulatory subunit M2 (RRM2) or insulin-like growth factor binding protein 2 (IGFBP-2), have shown efficient inhibition of pancreatic tumor growth and metastasis, and an increased chemotherapeutic sensitivity to conventional drugs such as doxorubicin or gemcitabine [62,63]. Yet, the simultaneous targeting of YAP and FOSL-1, a transcription factor that synergizes with YAP, showed the highest therapeutic effect in vivo, in agreement with the in vitro cytotoxicity. For that reason, a combined silencing of these two proteins could avoid the overexpression of several YAP-regulated genes, inducing a superior therapeutic effect.

In summary, our results open several possibilities for future studies to improve tumor targeting, finding a more efficient siRNA strategy and combining the silencing of FOSL-1 and YAP with other treatments, that could offer potential synergistic and complementary effects. Future work to translate this treatment into the clinic should include biodistribution studies and an orthotopic PDAC model.

Regarding their potential clinical utility and taking into account their biocompatibility in vitro and in vivo, we visualize two scenarios for the use of lipoplexes as potential therapy in patients with PDAC: (1) a directed targeting to the tumor by intravenous injection (in this case lipoplexes could require surface modification to be able to reach the tumor) and (2) local administration in the area after the surgical tumor removal (partial or complete).

## 4. Conclusions

Our study shows that the combined silencing of the transcription factor FOSL-1 and the transcriptional coactivator YAP by siRNA-lipoplexes exerts a therapeutic effect against pancreatic cancer, achieving a reduction in tumor development and growth on both immunocompetent and immunocompromised allografted mouse models. DODAB:MO (1:2) liposomes are effective nanocarriers for siRNA with a good cyto- and hemocompatibility profile and could work as adjuvant therapy, in combination with chemo-or/and immune-therapies. Likewise, the optimization of the silencing mechanism and targeting of the pancreatic tumor cells could help to increase their therapeutic efficiency. The combination of four different siRNAs, targeting the gene expression of FOSL-1 and YAP, paves the way for alternative therapeutic approaches on this lethal cancer.

## Figures and Tables

**Figure 1 cancers-14-03102-f001:**
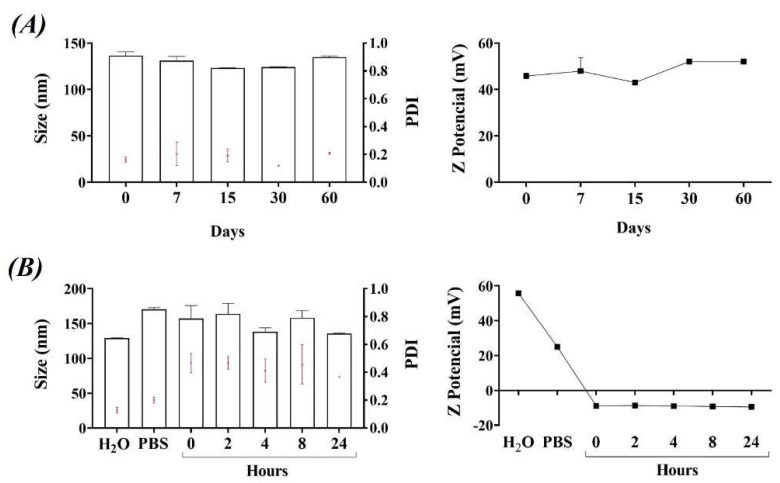
Stability characterization of DODAB:MO (1:2) liposomes. (**A**) Physicochemical characterization of liposomes in water at different time points (days) over two months and stored at 4 °C. (**B**) Physicochemical characterization of liposomes in water (H_2_O), PBS (PBS) or PBS + 10% FBS (kinetic study was performed at 0, 2, 4, 8 and 24 h). Z−averaged mean size (nm), polydispersity index (PDI), and ζ−potential (mV). The mean ± SD are represented. Size (bars), PDI (red symbols), and ζ-potential (lines and symbols). FBS: fetal bovine serum.

**Figure 2 cancers-14-03102-f002:**
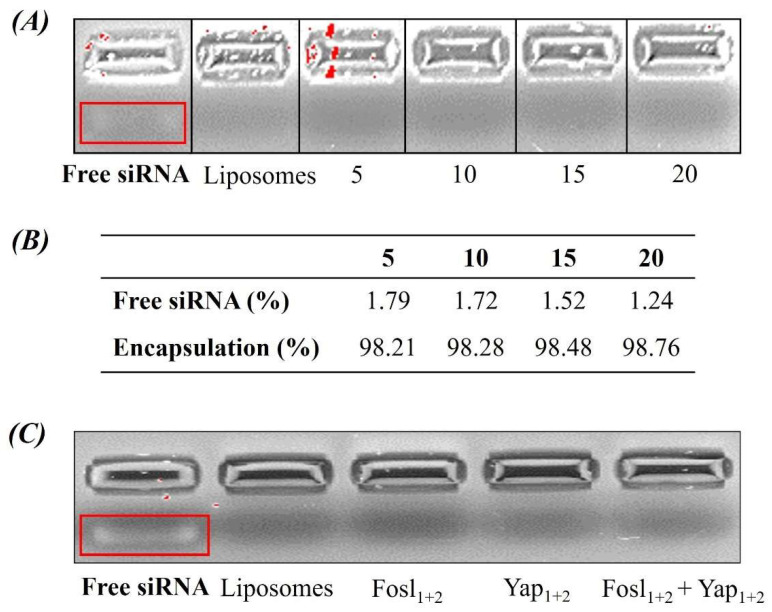
Determination of siRNA encapsulation. (**A**) Electrophoresis in agarose gel of lipoplexes formed using different charges ratios ± (5, 10, 15 and 20). (**B**) Quantification of non-encapsulated siRNA by using a fluorescence RNA label and encapsulation efficiency of the lipoplexes at different charge ratios (±). (**C**) Electrophoresis in agarose gel of lipoplexes using siRNA at charge ratio ± 15. Free siRNA and liposomes at 0.88 µM and 1.67 mM, respectively (the equivalent concentration in the lipoplexes), were used as controls.

**Figure 3 cancers-14-03102-f003:**
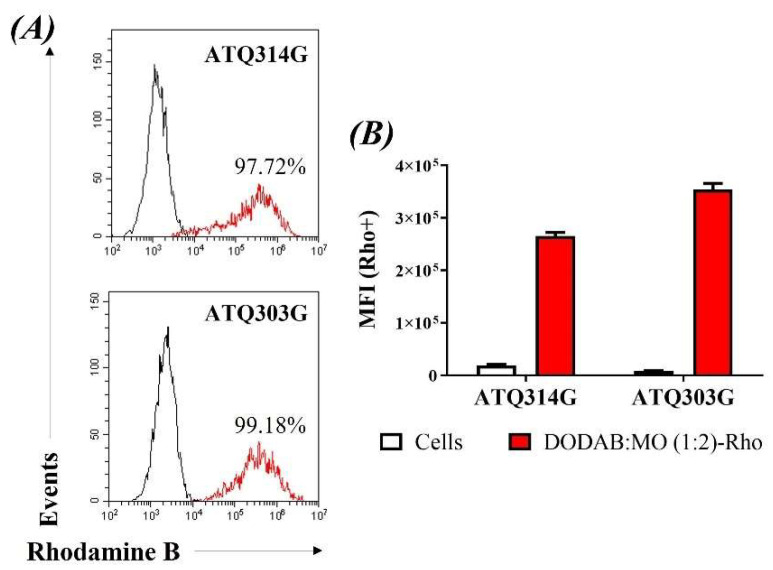
Cellular internalization of rhodamine labelled liposomes by the ATQ314G and ATQ303G cell lines after 5 h of incubation. (**A**) histograms or (**B**) median fluorescence intensity (MFI).

**Figure 4 cancers-14-03102-f004:**
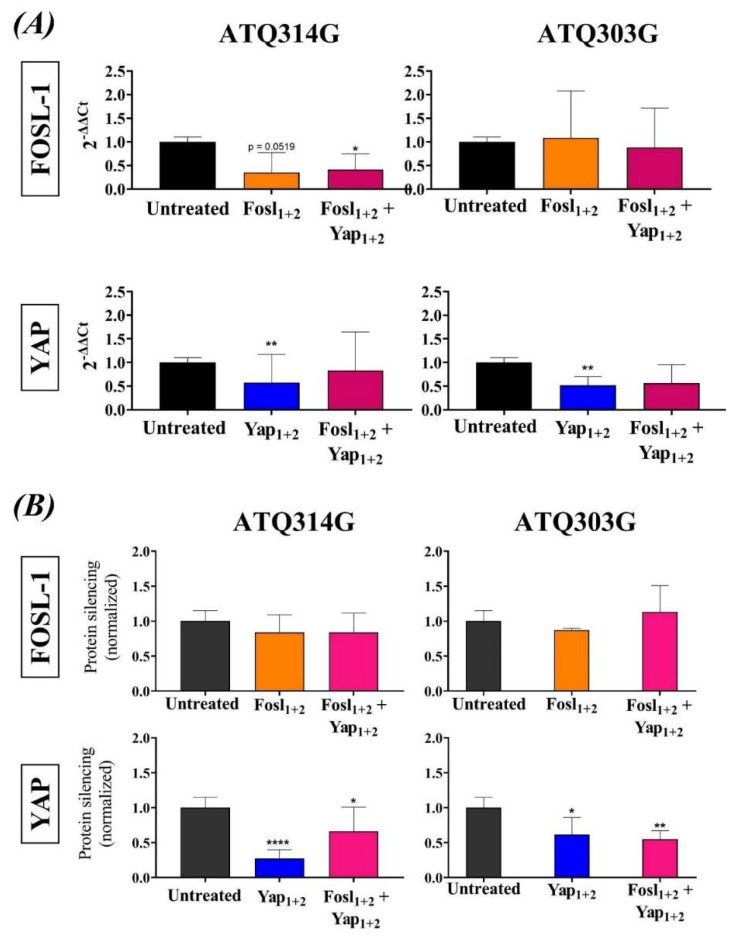
In vitro silencing efficacy of lipoplexes in ATQ314G and ATQ303G cells after 72 h of incubation with the different lipoplexes at charge ratio (±) 15 and 50 nM siRNA. (**A**) mRNA expression analysed by qRT−PCR of FOSL-1 and YAP using GAPDH as internal control and normalized to untreated cells. (**B**) Quantification of protein expression studied by Western blot using tubulin as control and normalized to untreated cells. In the graphs, the statistically significant differences between negative control and the different treatments were represented as: * *p* ≤ 0.05, ** *p* ≤ 0.01; **** *p* ≤ 0.0001. For original Western Blot, see Appendix A.

**Figure 5 cancers-14-03102-f005:**
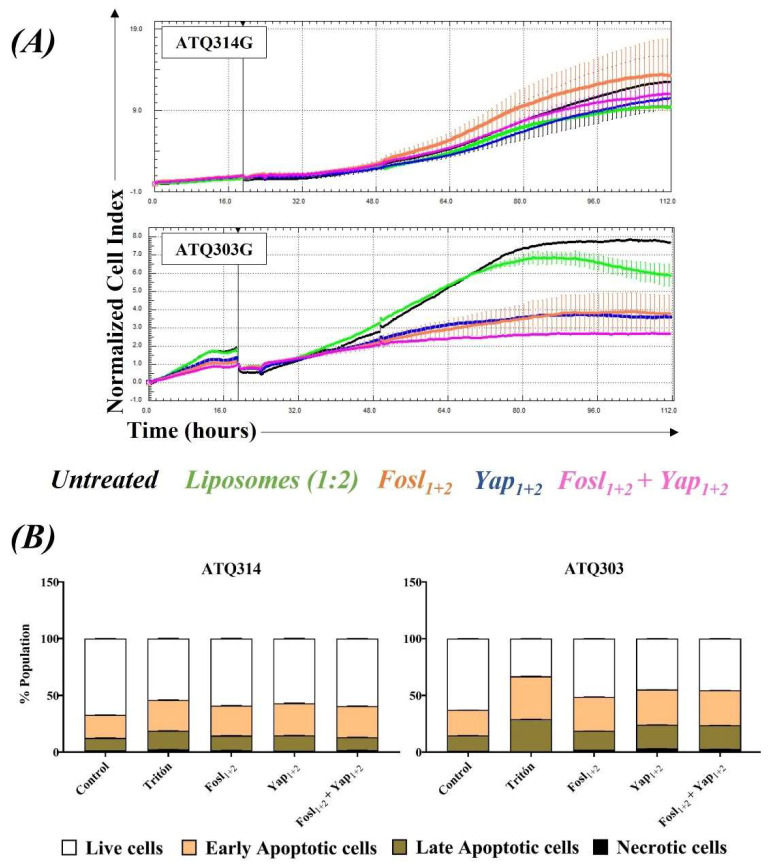
Cytotoxicity of lipoplexes in ATQ314G and ATQ303G cells. (**A**) Kinetics of the normalized cell index of the ATQ314G and ATQ303G cells during 96 h of incubation. (**B**) Percentage of different cell populations (live, early apoptotic, late apoptotic and necrotic cells) after 72 h of incubation determined by flow cytometry. Lipoplexes were tested at 50 nM siRNA and liposomes alone were tested at the equivalent lipid concentration (100 µM). Culture medium was used as negative control.

**Figure 6 cancers-14-03102-f006:**
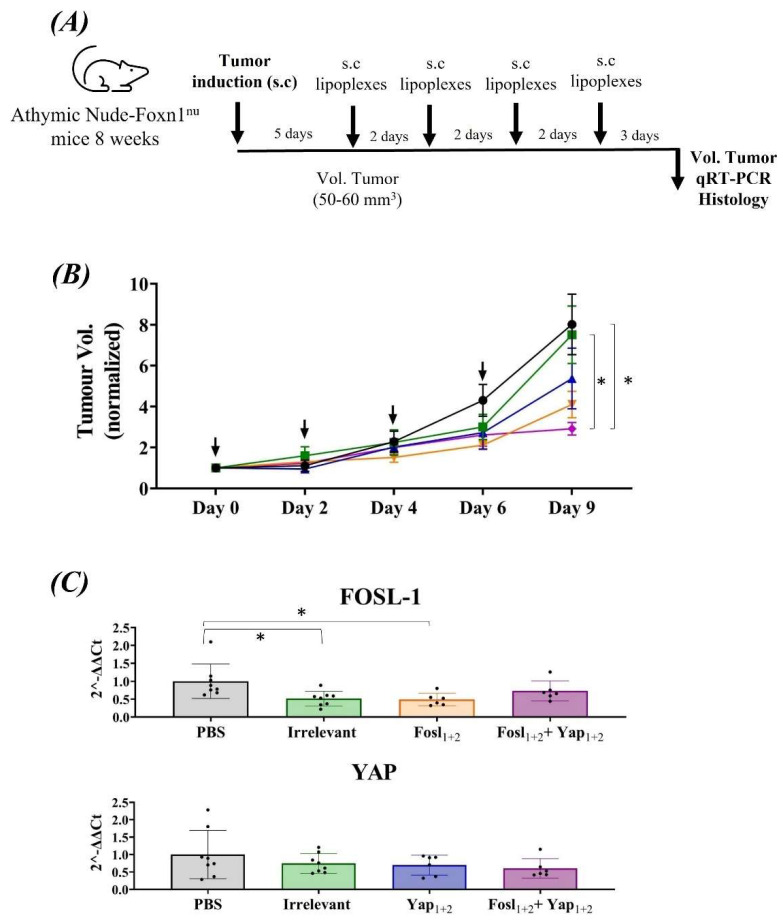
Anti-tumoral efficacy of lipoplexes in ATQ303G cell−allografted Athymic Nude−Foxn1^nu^ mice (*n* = 8). (**A**) Schedule of the study, including subcutaneous tumor cells inoculation and subsequent subcutaneous injection of the lipoplexes around the tumor. (**B**) Tumor size was normalized at the first day of drug administration (5 days after tumor cells inoculation). (**C**) mRNA expression of FOSL-1 and YAP after 72h of the last lipoplexes administration. The differences between PBS and the treatments were represented as: * *p* ≤ 0.05; while between irrelevant lipoplexes and treatments were represented in brackets (*) *p* ≤ 0.05. PBS (black), Irrelevant (green), Fosl_1+2_ (brown), Yap_1+2_ (blue) and combination of Fosl_1+2_ + Yap_1+2_ (purple).

**Figure 7 cancers-14-03102-f007:**
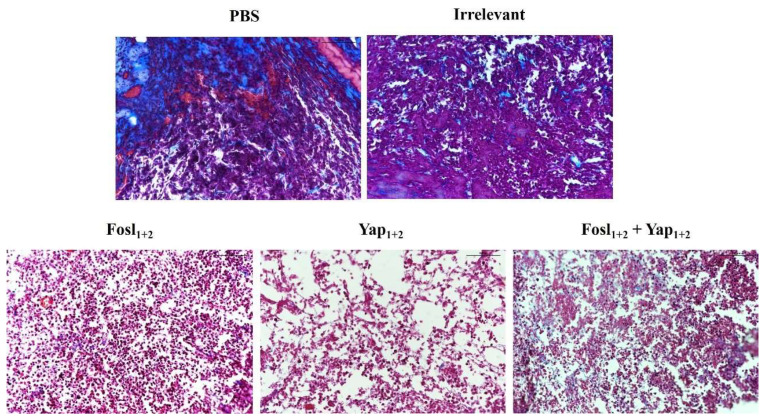
Masson’s trichrome stained paraffin sections of representative mouse tumors for each treatment (PBS, Irrelevant siRNA, Fosl_1+2_, Yap_1+2_ and combination of Fosl_1+2_ + Yap_1+2_ lipoplexes). Staining: blue (collagen), pink-purple (cytoplasm), red (muscle/erythrocytes) and black (cell nucleus). Scale bars: 100 µm. Magnification: 20×.

**Figure 8 cancers-14-03102-f008:**
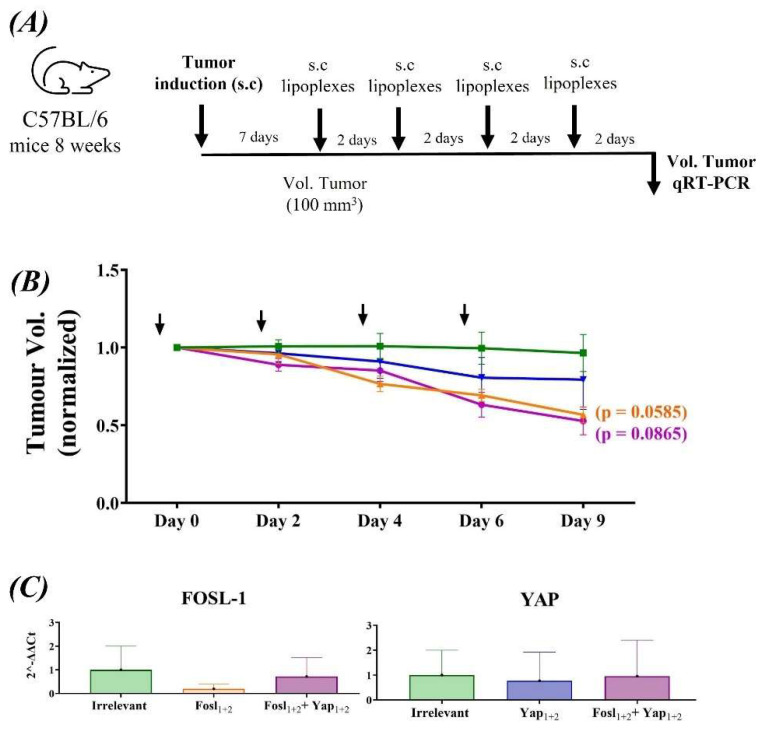
Anti-tumoral efficacy of lipoplexes in ATQ303G cell−allografted C57BL/6 mice (*n* = 8). (**A**) Schedule of tumor cells inoculation and subsequent injection of the lipoplexes subcutaneously around the tumor. (**B**) Tumor volume was normalized at the first day of drug administration (7 days after tumor induction). (**C**) mRNA expression of FOSL-1 and YAP after 48 h of the last lipoplexes administration. Irrelevant (green), Fosl_1+2_ (brown), Yap_1+2_ (blue) and combination of Fosl_1+2_ + Yap_1+2_ (purple). For the statistical analysis, animals treated with Yap_1+2_, Fosl_1+2_ and Fosl_1+2_ + Yap_1+2_ were compared to those treated with the irrelevant siRNA lipoplexes.

**Table 1 cancers-14-03102-t001:** Nucleotide sequences of the different siRNAs used for the inhibition of the FOSL-1 and YAP mRNA and protein expression, and the primers used for qRT-PCR.

siRNA	*5′–3′ Strand*	*3′–5′ Strand*
*Irrelevant*	GCAAACCACCAAUCUAACA	CGUUUGGUGGUUAGAUUGU
*Fosl1*	GGGCAGCUGCUAUUUAUUUUU	UUCCCGUCGACGAUAAAUAAA
*Fosl2*	GGUGCCCUUUGACUAGCCUTT	TTCCACGGGAAACUGAUCGGA
*Yap1*	GCUUUCUCACGUGGUUAUAUU	UUCGAAAGAGUGCACCAAUAU
*Yap2*	CCAAGCUAGAUAAAGAAAGTT	GTGGUUCGAUCUAUUUCUUUC
**Primers**	**Forward (Fw)**	**Reverse (Rv)**
*GAPDH*	5′-CCTCACCACCATGGAGGAGGC-3′	5′-GGCATGGACTGTGGTCATGAG-3′
*FOSL-1*	5′-ATGTACCGAGACTACGGGGAA-3′	5′-CTGCTGCTGTCGATGCTTG-3′
*YAP*	5′-GGATGTCTCAGGAATTGAGAACA-3′	5′-ATGCTGTAGCTGCTCATGCTGA-3′

**Table 2 cancers-14-03102-t002:** Physicochemical characterization of DODAB:MO (1:2) liposomes at 3 mM and lipoplexes at charge ratio (±) 15 containing 2 or 4 siRNAs.

	Size (nm)	PDI	ζ-Potential (mV)
Liposomes	139.0 ± 17.12	0.170 ± 0.03	(+) 50.7 ± 5.70
Lipoplexes (Fosl_1+2_)	129.6 ± 12.50	0.200 ± 0.02	(+) 54.0 ± 0.28
Lipoplexes (Yap_1+2_)	136.0 ± 13.20	0.100 ± 0.05	(+) 41.3 ± 0.85
Lipoplexes (Fosl_1+2_ + Yap_1+2_)	136.7 ± 13.60	0.160 ± 0.02	(+) 39.6 ± 12.3

Z-averaged mean size (nm), polydispersity index (PDI), and ζ-potential (mV). The mean ± standard deviation (SD) are represented.

## Data Availability

Data was included in the manuscript or in Appendix A.

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
