# Peer review of "Combined Inhibition of FOSL-1 and YAP Using siRNA-Lipoplexes Reduces the Growth of Pancreatic Tumor"

_cancers, 2022, doi:10.3390/cancers14133102_

Round 1

Reviewer 1 Report

The manuscript titled: “Combined inhibition of FOSL-1 and YAP using siRNA-lipoplexes reduces the growth of pancreatic tumor” is well-written. The manuscript lacks the molecular mechanism to support in vivo data observations. Data needs to be refined in a presentable manner. The present manuscript would benefit by addressing the points below.

In Figures 6 and 8, the authors showed expression of FOSL-1 and YAP. Along with this, the authors need to show knockdown of FOSL-1 and YAP in the tumor samples with western blot. According to figure 6 data, authors claim that knockdown of FOSL-1 and YAP suppresses tumor growth. Along with the tumor growth curve data, the authors should show the effect of inhibition of FOSL-1 and YAP on cell growth by immunohistochemistry of ki-67 and caspase 3 staining in the tumor sections.

2.    Authors showed the inhibition of FOSL1 and YAP hamper the tumor growth of ATQ303G cells in vivo. To support this claim, the authors need to show the knockdown of FOSL-1 and YAP affects the proliferation or cell death of ATQ303G cells in vitro.

3.    Since we see the knockdown of FOSL-1 and YAP affects the tumor growth, I request authors show this phenotype with FOSL-1 and YAP inhibitors.

4.    Further, the manuscript lacks the molecular mechanism behind FOSL-1 and YAP that involves regulating tumor growth. I'd like for the authors to do more work on this. 

Reviewer 2 Report

The manuscript “Combined inhibition of FOSL-1 and YAP using siRNA-lipoplexes reduces the growth of pancreatic tumor” by Dr Lara Diego-González describes the new liposome siRNA delivery system aiming to target FOSL-1 and YAP signaling pathway in therapy for PDAC. The new lipoplexes are well defined and characterized. The authors provide strong and detailed evidence to demonstrate the high efficiency of new cationic liposome mediated transfection of mouse pancreatic cancer cells. siRNA targeting FOSL-1 and YPA expression in both mRNA and protein levels is approved in in vitro and in vivo studies. However, the idea using cationic liposome delivering siRNA therapy was tried 20 years ago.  The targeting Yap and FosL is not new strategy. The subQ xenograft animal models do not recapitulate real tumor environment in pancreas, which should be avoided. Near tumor injection of therapeutic siRNA is not practicable in treatment of PDAC in reality. Ideally, the RNAi therapies should use as few type of siRNA as possible to avoid possible off-targets. There is no studies of off-target in this manuscript. 

Fig 2A, the bands look too weak to be measurable

Fig 2B, it would be better to show the saturation charge ratio

Is there EM imaging to show the size and morphology of lipoplexes or aggregation.

Line 443  “FOSL-1 mRNA level was also significantly inhibited in the ATQ314G cell line, but not in ATQ303G cells”. P value doesn’t show here. On Figure 4A, a p=0.0519 showing above bar FosL is unclear to which it compares, which means no significant. It needs to clarify which treatment(s) lead(s) to significant inhibition of FOSL-1 mRNA.

In Figure 4B, it shows no significant inhibition of FosL-1 protein expression “Down-regulation was also observed for the FOSL-1 protein but did not reach statistical significance.” Is it transfection problem or siRNA not working? Please clarify. A positive liposome control such as lipofectamine should be set aiming to compared to homemade liposome in terms of transfection efficiency. 

The western band patterns should be integrated into Fig 4B. It is not clear what the “another treatment” means in supplemental figure.

There is no biodistribution data to show where the lipoplexes go

In figure 6C, please make clarification of which groups have significant differences. Using line indication would be better.

In figure 6C FOSL-1 expression, it looks like no significance between irrelevant vs FosI1+2, however, irrelevant treatment causes inhibition of FOSL1. 

The Figure 7 has a bad quality of imaging. It doesn’t show the structure clearly. Greater magnification of microscopic imaging is needed. I can’t see the scale bar

It is not very understandable on Figure 8B where it looks like that the control group (irrelevant) also causes suppression of tumor growing. 

In Figure 8C, no significance is indicated, 

The conclusion needs to be more clear and specific. “These results open a new perspective on the therapeutic use of lipoplexes for pancreatic cancer” means nothing. The liposomal delivery of RNAi therapy for PDAC was tried 20 years ago. The authors should address what the novelty in this studies as compared to previous studied liposome RNAi therapy.

Round 2

Reviewer 1 Report

The authors have addressed all comments. The manuscript is improvised after revision, and the present version of the manuscript is in acceptable form.

Reviewer 2 Report

The authors address all the questions and comments.